

**Predominance of methanogens over methanotrophs contributes**
**to high methane emissions in rewetted fens**
Xi Wen[1*], Viktoria Unger[2*], Gerald Jurasinski[2], Franziska Koebsch[2], Fabian Horn[1], Gregor
Rehder[3], Torsten Sachs[4], Dominik Zak[5,6], Gunnar Lischeid[7,8], Klaus-Holger Knorr[9], Michael
Böttcher[10], Matthias Winkel[1], and Susanne Liebner[1,11].
[1]Section 5.3 Geomicrobiology, GFZ German Research Centre for Geosciences, Helmholtz Centre
Potsdam, Telegrafenberg, Potsdam, 14473, Germany
[2]Landscape Ecology and Site Evaluation, Faculty for Agricultural and Environmental Sciences,
Rostock University, Rostock, 18059, Germany
[3]Department of Marine Chemistry, Leibniz Institute for Baltic Sea Research, Warnemünde, 18119,
Germany
[4]Section 1.4 Remote Sensing, GFZ German Research Centre for Geosciences, Helmholtz Centre
Potsdam, Telegrafenberg, Potsdam, 14473, Germany
[5]Department of Bioscience, Aarhus University, Silkeborg, 8600, Denmark
[6]Department of Chemical Analytics and Biogeochemistry, Leibniz Institute of Freshwater Ecology
and Inland Fisheries, Berlin, 12587, Germany
[7]Institute of Landscape Hydrology, Leibniz Center for Agricultural Landscape Research,
Münchberg, 15374, Germany
[8]Institute of Earth and Environmental Science, University of Potsdam, Potsdam, 14476, Germany
[9]Institute of Landscape Ecology, University of Münster, Münster, 48149, Germany
[10]Geochemistry and Stable Isotope Biogeochemistry, Leibniz Institute for Baltic Sea Research,
Warnemünde, 18119, Germany
[11]University of Potsdam, Institute of Biochemistry and Biology, Potsdam, 14469, Germany
*Correspondence to*: Viktoria Unger (viktoria.unger@uni-rostock.de), Franziska Koebsch
(franziska.koebsch@uni-rostock.de)
*Shared first authorship – the two first authors contributed equally to preparation of this work
**Abstract.** The rewetting of drained peatlands alters peat geochemistry and often leads to sustained
elevated methane emission. Although this methane is produced entirely by microbial activity, the
distribution and abundance of methane-cycling microbes in rewetted peatlands, especially in fens,
is rarely described. In this study, we compare the community composition and abundance of
methane-cycling microbes in relation to peat porewater geochemistry in two rewetted fens in
northeastern Germany, a coastal brackish fen and a freshwater riparian fen, with known high
methane fluxes. We utilized 16S rDNA high-throughput sequencing and quantitative polymerase



chain reaction on 16S rDNA, *mcrA*, and *pmoA* genes to determine microbial community
composition and the abundance of total bacteria, methanogens, and methanotrophs. Electrical
conductivity was more than three times higher in the coastal fen than in the riparian fen, averaging
5.3 and 1.5 mS cm$^{-1}$, respectively. Porewater concentrations of terminal electron acceptors varied
within and among the fens. This was also reflected in similarly high intra- and inter-site variations
of microbial community composition. Despite these differences in environmental conditions and
electron acceptor availability, we found a low abundance of methanotrophs and a high abundance
of methanogens, represented in particular by *Methanosaetaceae*, in both fens. This suggests that
rapid re/establishment of methanogens and slow re/establishment of methanotrophs contributes to
prolonged increased methane emissions following rewetting.

## 44 1 Introduction

Rewetting is a technique commonly employed to restore ecological and biogeochemical
functioning of drained fens. However, while rewetting may reduce carbon dioxide ($CO_2$) emissions
(Wilson et al. 2016), it often increases methane ($CH_4$) emissions in peatlands that remain mostly
inundated following rewetting. The factors that contribute to the magnitude and duration of this
increase are still uncertain (Joosten et al. 2015, Abdalla et al. 2016). On a 100-year time scale $CH_4$
has a global warming potential 28 times stronger than $CO_2$ (Myhre et al. 2013); thus, increased
$CH_4$ emissions could potentially offset the benefit of decreased $CO_2$ emissions (Jurasinski et al.
2016). Although a recent increase in rewetting projects in Germany and other European nations
has prompted a number of studies of methane cycling in rewetted peatlands (e.g., Jerman et al.
2009, Hahn-Schöfl et al. 2011, Urbanová et al. 2013, Hahn et al. 2015, Vanselow-Algan et al.
2015, Zak et al. 2015, Emsens et al. 2016), the post-rewetting distribution and abundance of
methane-cycling microbes in rewetted fens has seldom been examined (but see Juottonen et al.
2012, Urbanová et al. 2013).
Peat $CH_4$ production and release is governed by a complex array of interrelated factors including
climate, water level, plant community, nutrient status, site geochemistry, and the activity of
microbes (i.e. bacteria and archaea) that use organic carbon as energy source (Segers 1998, Abdalla
et al. 2016). To date, the vast majority of studies in rewetted fens have focused on quantifying $CH_4$



emission rates in association with environmental variables such as water level, plant community,
and aspects of site geochemistry (Abdalla et al. 2016). Site geochemistry indeed plays an important
role for methanogenic communities, as methanogenesis is suppressed in presence of
thermodynamically more favorable terminal electron acceptors (TEAs, Blodau 2011). Due to a
smaller pool of more favorable electron acceptors and high availability of carbon substrates,
organic-rich soils such as peat rapidly establish methanogenic conditions when anoxic (Segers
1998, Keller and Bridgham 2007, Knorr and Blodau 2009). Despite their decisive role as producers
(i.e. methanogens) and consumers (i.e. methanotrophs) of $CH_4$ (Conrad 1996), only a few studies
have combined a characterization of the $CH_4$-cycling microbial community, site geochemistry, and
observed patterns of $CH_4$ production. Existing studies have been conducted in oligotrophic and
mesotrophic boreal fens (e.g., Juottonen et al. 2005, Yrjälä et al. 2011, Juottonen et al. 2012),
alpine fens (e.g., Liebner et al. 2012, Urbanová et al. 2013, Cheema et al. 2015, Franchini et al.
2015), subarctic fens (Liebner et al. 2015), and incubation experiments (e.g., Jerman et al. 2009,
Knorr and Blodau 2009, Urbanová et al. 2011, Emsens et al. 2016). Several studies on $CH_4$-cycling
microbial communities have been conducted in minerotrophic temperate fens (e.g., Cadillo-Quiroz
et al. 2008, Liu et al. 2011, Sun et al. 2012, Zhou et al. 2017), but these sites were not subject to
drainage or rewetting. To our knowledge, only one study has directly compared *in situ* abundances
of methanogens and methanotrophs in drained versus rewetted fens (Juottonen et al. 2012). The
studied sites, however, were nutrient-poor fens with acidic conditions.
While studies of nutrient-poor and mesotrophic boreal fens have documented post-rewetting $CH_4$
emissions comparable to or lower than at pristine sites (Komulainen et al. 1998, Tuittila et al. 2000,
Juottonen et al 2012), studies of temperate nutrient-rich fens have reported post-flooding $CH_4$
emissions dramatically exceeding emissions in pristine fens (e.g., Augustin and Chojnicki 2008,
Hahn et al. 2015). These high emissions typically occur together with a significant dieback in
vegetation, a mobilization of nutrients and electron acceptors in the upper peat layer, and increased
availability of dissolved organic matter (Zak and Gelbrecht 2007, Hahn-Schöfl et al. 2011, Hahn



et al. 2015, Jurasinski et al. 2016). Vanselow-Algan et al. (2015) have shown that such high $CH_4$
fluxes may continue for decades following rewetting even in bogs. Because of their potential to
remain significant $CH_4$ sources on decadal timescales, there is an urgent need to characterize $CH_4$-
cycling microbial communities and geochemical conditions in rewetted minerotrophic fens.
Therefore, in this study, we examined microbial community composition and abundance in
relation to post-flooding geochemical conditions in two rewetted fens in northeastern Germany. In
both fens, $CH_4$ emissions increased dramatically after rewetting (Augustin and Chojnicki 2008,
Hahn-Schöfl et al. 2011, Hahn et al. 2015, Jurasinski et al. 2016). Average annual $CH_4$ emissions
have decreased in both fens since the initial peak (Franz et al. 2016, Jurasinski et al. 2016).
Nevertheless, fluxes remained higher than under pre-flooding conditions (ibid.), and higher than
in pristine fens (Urbanová et al. 2013, Minke et al 2016).
We expected patterns in microbial community composition would reflect the geochemical
conditions of the two sites and hypothesized a high abundance of methanogens relative to
methanotrophs in both fens. We also expected acetoclastic methanogens, which typically thrive in
nutrient-rich fens (Kelly et al. 1992, Galand 2005), to dominate the methanogenic community in
both fens.

**2 Methods**
**2.1 Study sites**
The nature reserve "Heiligensee and Hütelmoor" ('Hütelmoor' in the following, approx. 540 ha,
54°12'36.66" N, 12°10'34.28" E), is a coastal, mainly minerotrophic fen complex in Mecklenburg-
Vorpommern (NE Germany) that is separated from the Baltic Sea by a narrow (~100 m and less)
dune dike (Fig. 1a and b). The climate is temperate in the transition zone between maritime and
continental with an average annual temperature of 9.1 °C and an average annual precipitation of
645 mm (data derived from grid product of the German Weather Service, reference climate period:



1981–2010). Episodic flooding from storm events delivers sediment and brackish water to the site
(Weisner and Schernewski 2013). The vegetation is a mixture of salt-tolerant macrophytes, with
dominant to semi-dominant stands of *Phragmites australis*, *Bolboschoenus maritimus*, *Carex*
*acutiformis*, and *Schoenoplectus tabernaemontani.* The dominating plants are interspersed with
open water bodies that are colonized by *Ceratophyllum demersum* in summer (Koch et al. 2017).
Intense draining and land amelioration practices began in the 1970s, which lowered the water level
to 1.6 m below ground surface and caused aerobic decomposition and concomitant degradation of
the peat (Voigtländer et al. 1996). The upper peat layer varies in depth between 0.6 and 3 m and
is highly degraded, reaching up to H10 on the von Post humification scale (Hahn et al. 2015).
Active draining ended in 1992, but dry conditions during summertime kept the water table well
below ground surface (Schönfeld-Bockholt et al. 2005, Koebsch et al. 2013) until concerns of
prolonged aerobic peat decomposition prompted the installation of a weir in 2009 at the outflow
of the catchment (Weisner and Schernewski 2013). After installation of the weir, the site was fully
flooded year-round with an average water level of 0.6 m, and annual average $CH_4$ flux increased
~186-fold from $0.0014 \pm 0.0006$ kg $CH_4$ m$^{-2}$ a$^{-1}$ to $0.26 \pm 0.06$ kg $CH_4$ m$^{-2}$ a$^{-1}$ (Hahn et al. 2015).
The study site polder Zarnekow ('Zarnekow' in the following, approx. 500 ha, 53°52'31.10" N,
12°53'19.60" E) is situated in the valley of the River Peene in Mecklenburg-Vorpommern (NE
Germany, Fig. 1a and c). The climate is slightly more continental compared to the Hütelmoor, with
a mean annual precipitation of 544 mm and a mean annual temperature of 8.7 °C (German Weather
Service, meteorological station Teterow, 24 km southwest of the study site; reference period 1981–
2010). The fen can be classified as a river valley mire system consisting of spring mires, wider
percolation mires, and flood mires along the River Peene. Drainage and low-intensity agricultural
use began in the eighteenth century when land-use changed to pastures and grassland. This was
intensified by active pumping in the mid-1970s. Due to land subsidence of several decimeters,
after rewetting (October 2004) water table depth increased to 0.1–0.5 m above peat surface. The
upper horizon is highly decomposed (0–0.3 m), followed by moderately decomposed peat to a





depth of 1 m and a deep layer of slightly decomposed peat up to a maximum depth of 10 m. The
open water bodies are densely colonized by *Ceratophyllum* spp. and *Typha latifolia* is the dominant
emergent macrophyte (Steffenhagen et al. 2012). Following flooding, $CH_4$ flux rates increased to
~0.21 kg m$^{-2}$ a$^{-1}$ (Augustin and Chojnicki 2008). No pre-rewetting $CH_4$ flux data were available
for the Zarnekow site but published $CH_4$ flux rates of representative drained fens from the same
region have been shown to be negligible (Augustin et al. 1998).
**2.2 Collection of peat cores and porewater samples**
Peat and porewater samples were collected at four different locations in Hütelmoor (October 2014)
and at five locations in Zarnekow (July 2015) and spanned a distance of 1,200 m and 250 m,
respectively, to cover the whole lateral extension at each site (Fig. 1b and c). Peat cores were
collected with a Perspex liner (ID: 60 mm, Hütelmoor) and a peat auger (Zarnekow). In order to
minimize oxygen contamination, the outer layer of the peat core was omitted. Subsamples for
molecular analysis were immediately packed in 50 ml sterile Falcon tubes and stored at -80 °C
until further processing.
Pore waters in Hütelmoor were collected with a stainless-steel push-point sampler attached to a
plastic syringe to recover the samples from 10 cm depth intervals. Samples were immediately
filtered with 0.45 μm membrane disposable syringe filters. Pore waters in Zarnekow were sampled
with permanently installed dialysis samplers consisting of slotted polypropylene (PP) pipes
(length: 636 mm, ID: 34 mm) surrounded with 0.22 μm polyethersulfone membrane. The PP pipes
were fixed at distinct peat depths (surface level, 20 and 40 cm depth) and connected with PP tubes
(4x6 mm IDxAD). Water samples were drawn out from the dialysis sampler pipes with a syringe
through the PP tube.
At both sites, electrical conductivity (EC), dissolved oxygen (DO) and pH were measured
immediately after sampling (Sentix 41 pH probe and a TetraCon 325 conductivity measuring cell
attached to a WTW multi 340i handheld; WTW, Weilheim). Headspace $CH_4$ concentrations of
porewater samples were measured with an Agilent 7890A gas chromatograph (Agilent



Technologies, Germany) equipped with a flame ionization detector and a Carboxen PLOT
Capillary Column or HP-Plot Q (Porapak-Q) column. The measured headspace $CH_4$ concentration
was then converted into a dissolved $CH_4$ concentration using the temperature-corrected solubility
coefficient (Wilhelm et al. 1977). Isotopic composition of dissolved $CH_4$ for Hütelmoor was
analyzed using the gas chromatography-combustion-technique (GC-C) and the gas
chromatography-high-temperature-conversion-technique (GC-HTC). The gas was directly
injected in a Gas Chromatograph Agilent 7890A, methane was quantitatively converted to $CO_2$
and the $\delta^{13}C$ values were then measured with the isotope-ratio-mass-spectrometer MAT-253
(Thermo Finnigan, Germany). The $\delta^{13}C$ of dissolved methane in Zarnekow was analyzed using a
laser-based isotope analyzer equipped with a small sample isotope module for analyses of discrete
gas samples (cavity ring down spectroscopy CRDS; Picarro G2201-I, Santa Clara, CA, USA).
Calibration was carried out before, during and after analyses using certified standards of known
isotopic composition (obtained from Isometric Instruments, Victoria, BC, Canada, and from
Westfalen AG, Münster, Germany). Reproducibility of results was typically +/- 1 ‰. In the
presence of high concentrations of hydrogen sulfide interfering with laser-based isotope analysis,
samples were treated with iron(III) sulfate to oxidize and/or precipitate sulfide. For both sites,
sulfate and nitrate concentrations were analyzed by ion chromatography (IC, Thermo Fisher
Scientific Dionex) using an Ion Pac AS-9-HC 4 column, partly after dilution of the sample.
Dissolved metal concentrations were analyzed by ICP-OES (iCAP 6300 DUO, Thermo Fisher
Scientific). Accuracy and precision were routinely checked with a certified CASS standard as
previously described (Kowalski et al. 2012).
**2.3 Gene amplification and phylogenetic analysis**
Genomic DNA was extracted from 0.2–0.3 g of duplicates of peat soil per sample using an EurX
Soil DNA Kit (Roboklon, Berlin, Germany). DNA concentrations were quantified with a
Nanophotometer P360 (Implen GmbH, München, DE) and Qubit 2.0 Fluorometer (Thermo Fisher
Scientific, Darmstadt, Germany). Polymerase chain reaction (PCR) amplification of bacterial and





archaeal 16S rRNA genes was performed using the primer combination of S-D-Bact-0341-b-S-
17/S-D-Bact-0785-a-A-21 (Herlemann et al. 2011) and S-D-Arch-0349-a-S-17/S-D-Arch-0786-a-
A-20 (Takai and Horikoshi 2000), respectively. The PCR mix contained 1x PCR buffer (Tris•Cl,
KCl, $(NH_4)_2SO_4$, 15 mM $MgCl_2$; pH 8.7) (QIAGEN, Hilden, Germany), 0.5 µM of each primer
(Biomers, Ulm, Germany), 0.2 mM of each deoxynucleoside (Thermo Fisher Scientific,
Darmstadt, Germany) and 0.025 U µl$^{-1}$ hot start polymerase (QIAGEN, Hilden, Germany). PCR
samples were kept at 95 °C for 5 min to denature the DNA, with amplification proceeding for 40
cycles at 95 °C for 1 min, 56 °C for 45 s and 72 °C for 90 s; a final extension of 10 min at 72 °C
was added to ensure complete amplification. PCR products were purified with a Hi Yield Gel/PCR
DNA fragment extraction kit (Süd-Laborbedarf, Gauting, Germany). PCR products of three
individual runs per sample were combined. PCR products of different samples were pooled in
equimolar concentrations and compressed to a final volume of 10 µl with a concentration of 200
ng µl$^{-1}$ in a vacuum centrifuge Concentrator Plus (Eppendorf, Hamburg, Germany).
Illumina sequencing was performed by GATC Biotech AG using 300 bp paired-end mode and a
20% PhiX Control v3 library to counteract the effects of low-diversity sequence libraries. Raw
data was demultiplexed using an own script based on CutAdapt (Martin 2011). Ambiguous
nucleotides at sequence ends were trimmed and a 10% mismatch was allowed for primer
identification, whereas barcode sequences needed to be present without any mismatches and with
a minimum Phred-Score of Q25 for each nucleotide. After sorting, overlapping paired-end reads
were merged using PEAR [Q25, p 0.0001, v20] (Zhang et al. 2014). The orientation of the merged
sequences was standardized according to the barcode information obtained from demultiplexing.
Low-quality reads were removed using Trimmomatic [SE, LEADING Q25, TRAILING Q25,
SLIDINGWINDOW 5:25; MINLEN 200] (Bolger et al. 2014). Chimeric sequences were removed
using USEARCH 6.1 and the QIIME-script identify_chimeric_seqs.py (Caporaso et al. 2010). Pre-
processed sequences were taxonomically assigned to operational taxonomic units (OTUs) at a
nucleotide sequence identity of 97% using QIIME's pick_open_reference_otus.py script and the





GreenGenes database 13.05 (McDonald et al. 2012) as reference. The taxonomic assignment of
representative sequences was further checked for correct taxonomical classification by
phylogenetic tree calculations in the ARB environment referenced against the SILVA database
(https://www.arb-silva.de) version 119 (Quast et al. 2013). The resulting OTU table was filtered
for singletons, OTUs assigned to chloroplasts or mitochondria, and for low-abundance OTUs
(below 0.2% within each sample). Archaeal and bacterial samples were processed separately while
only OTUs that were assigned to the respective domain were considered for further analysis. The
16S rRNA gene sequence data have been deposited at NCBI under the Bioproject PRJNA356778.
Hütelmoor sequence read archive accession numbers are SRR5118134-SRR5118155 for bacterial
and SRR5119428-SRR5119449 for archaeal sequences, respectively. Zarnekow accession
numbers are SRR6854018-SRR6854033 and SRR6854205-SRR6854220 for bacterial and
archaeal sequences, respectively.
**2.4 qPCR analysis**
Quantitative polymerase chain reaction (qPCR) for the determination of methanotrophic and
methanogenic functional gene copy numbers and overall bacterial 16S rRNA gene copy numbers
was performed via SybrGreen assays on a Bio-Rad CFX instrument (Bio-Rad, Munich, Germany)
with slight modifications after Liebner et al. (2015). The functional methanotrophic *pmoA* gene
was amplified with the primer combination A189F/Mb661 (Kolb et al. 2003) suitable for detecting
all aerobic methanotrophic Proteobacteria. Annealing was done at 55 °C after a 7-cycle-step
touchdown starting at 62 °C. The functional methanogenic *mcrA* gene was amplified with the
*mlas/mcrA*-rev primer pair (Steinberg and Regan 2009) with annealing at 57 °C. The bacterial 16S
rRNA gene was quantified with the primers Eub341F/Eub534R according to Degelmann et al.
(2010) with annealing at 58 °C. Different DNA template concentrations were tested prior to the
qPCR runs to determine optimal template concentration without inhibitions through co-extracts.
The 25 µl reactions contained 12.5 µl of iTaq universal Sybr Green supermix (Bio-Rad, Munich,
Germany), 0.25 µM concentrations of the primers, and 5 µl of DNA template. Data acquisition




was always done at 80 °C to avoid quantification of primer dimers. The specificity of each run
was verified through melt-curve analysis and gel electrophoresis. Only runs with efficiencies
between 82 and 105% were used for further analysis. Measurements were performed in triplicates.
We determined the ratio of methanogens to methanotrophs based on gene abundances of *mcrA* and
*pmoA*. The marker gene for the soluble monooxygenase, *mmoX*, was neglected due to the absence
of *Methylocella* in the sequencing data (Fig. 3).
**2.5 Data visualization and statistical analysis**
All data visualization and statistical analysis were done in R (R Core Team). The taxonomic
relative abundances across samples were visualized through bubble plots with the R package
ggplot2 (Wickham 2009). Differences in microbial community composition were visualized with
2-dimensional non-metric multidimensional scaling (NMDS) based on Bray-Curtis distances. The
NMDS ordinations were constructed using R package vegan (Oksanen et al. 2017). An
environmental fit was performed on the ordinations to determine the measured geochemical
parameters that may influence community composition. The geochemical data were fitted to the
ordinations as vectors with a significance of $p < 0.05$. Depth profiles were constructed with the
porewater geochemical data, as well as with the microbial abundances, to elucidate depthwise
trends and assess whether differences in microbial community and abundances among the two fens
are related to differences in their respective geochemistry.

**3 Results**
**3.1 Community composition of bacteria and archaea**
Bacterial sequences could be affiliated into a total of 30 bacterial phyla (Fig. 2). Among them,
Proteobacteria, Acidobacteria, Actinobacteria, Chloroflexi, Nitrospirae and Bacteroidetes were
present in all samples. With mean relative abundance of 48%, Proteobacteria was the most
abundant phylum. Some taxa (e.g., Verrucomicrobia, Atribacteria (OP9), and AD3) were present
only in Hütelmoor. Variation in community composition was larger in Hütelmoor samples than in



Zarnekow. Within Proteobacteria, the alpha subdivision was the most dominant group, having
contributed 26.7% to all the libraries on average (Fig. 3). The family *Hyphomicrobiaceae*
dominated the Alphaproteobacteria, and was distributed evenly across samples, but missing in the
surface and bottom peat layers in Hütelmoor core (HC) 2. In addition, methanotrophs were clearly
in low abundance across all samples. Of the few methanotrophs that were detected, type II
methanotrophs (mainly *Methylocystaceae*) outcompeted type I methanotrophs (mainly
*Methylococcaceae*) in the community, while members of the genus *Methylocella* were absent (Fig.

276 3).

Within the archaeal community, Bathyarchaeota were mostly dominating over Euryarchaeota (Fig.
4). The MCG group (mainly the order of pGrfC26) in Bathyarchaeota prevailed across all samples
but was especially abundant in HC 2 samples. In addition to Bathyarchaeota, methanogenic
archaea were important, and on average contributed 30.6% to the whole archaeal community.
Among the methanogens, acetoclastic methanogens were more abundant in most of the samples
and *Methanosaetaceae* (24.8%) were the major component. They were present in most samples
and much more dominant than *Methanosarcinaceae* (2.0%). Hydrogenotrophic methanogens, such
as *Methanomassiliicoccaceae* (1.6%), *Methanoregulaceae* (1.2%) and *Methanocellaceae* (0.6%),
albeit low in abundance, were detected in many samples. Hütelmoor samples displayed greater
variability in archaeal community composition compared to Zarnekow samples. The putative
anaerobic methanotrophs of the ANME-2D (Raghoebarsing et al. 2006) clade occurred in patchy
abundance with dominance in single spots of both sites. In HC 1 they represented a mean relative
abundance of 40.9% of total archaeal reads but were almost absent in all other Hütelmoor cores.
In Zarnekow core (ZC) 3, ANME-2D represented up to approximately 30% of all archaea but were
otherwise low in abundance.
**3.2 Environmental characteristics and site geochemistry**
The two rewetted fens varied substantially in their environmental characteristics (e.g., proximity
to the sea) and porewater geochemistry (Fig. 5, Tables 1 and 2). Electrical conductivity was more



than three times higher in Hütelmoor than in Zarnekow, averaging 5.3 and 1.5 mS cm$^{-1}$,
respectively.   Mean pH was approximately neutral (6.5 to 7) in the upper peat profile and
comparable in both fens until a depth of about 30 cm where pH was ~6 in the Hütelmoor.
Concentrations of the TEAs nitrate and sulfate were lower in Zarnekow and near zero in the pore
water at all depths, while nitrate and sulfate were abundant in the upper and lower peat profile in
Hütelmoor at ~1.5 to 3.0 mM and ~4 to 20 mM, respectively (Fig. 5). Iron concentrations were
higher in the Hütelmoor pore water, while manganese concentrations were higher in Zarnekow
pore water. Dissolved oxygen concentrations in the upper peat profile (i.e. 0 to 25 cm depths) were
much higher in Hütelmoor than in Zarnekow (Fig. 5). Here DO concentrations averaged ~0.250
mM until a depth of 15 cm at which they dropped sharply, reaching concentrations slightly below
0.050 mM at 25 cm. In Zarnekow, DO concentrations did not exceed 0.1 mM and varied little with
depth. Regarding geochemical conditions, HC 1 was distinct from all other Hütelmoor cores and
more similar to Zarnekow cores. In HC 1 – the core taken nearest to potential freshwater sources
(Fig. 1b) – pore water EC and DO concentrations were lower while pH was slightly higher than
all other Hütelmoor cores. Moreover, this was the only Hütelmoor core where nitrate
concentrations were undetectable (Fig. 5). Dissolved CH$_4$ concentrations were high, varied within
and among fens and were slightly higher in Zarnekow pore water. Stable isotope ratios of $\partial^{13}$C-
CH$_4$ (Fig. 5) in the upper peat (approx. −59‰) suggest a predominance of acetoclastic
methanogenesis, with a shift to hydrogenotrophic methanogenesis around −65‰ in the lower peat
profile. Also, shifts toward less negative $\partial^{13}$C-CH$_4$ values in the upper peat layer, as in HC 1 and
HC 2, could indicate partial oxidation of CH$_4$ occurred (Chasar et al. 2000).
**3.3 Environmental drivers of microbial community composition**
Bacterial and archaeal population at both peatland sites showed distinct clustering (Fig. 6) with
similarly high intra- and inter-site variations but greater overall variation in community
composition in the Hütelmoor. Community composition varied much more strongly in HC 2 than
in any other core (grey dashed-line polygon in Fig. 6). Bacterial communities in HC 1 were more



similar to communities in all Zarnekow cores than in other Hütelmoor cores (Fig. 6a). The archaeal
community in HC 1 was more similar to Zarnekow cores as well (Fig. 6b). Overall, the influence
of depth on microbial community was evident, especially in the Hütelmoor where the differences
were more pronounced. Environmental fit vectors suggest pH, oxygen and alternative TEA
availability as important factors influencing microbial community composition. The EC vector
suggests the importance of brackish conditions in shaping microbial communities in the Hütelmoor
(Fig. 6a - c).
**3.4 Total microbial and functional gene abundances**
Quantitative PCR results show that in both fens, *mcrA* abundance is up to two orders of magnitude
greater than *pmoA* abundance (Fig. 7, Tables 1 and 2). Gene copy numbers of *mcrA* are overall
higher and spatially more stable in Zarnekow than in Hütelmoor. Total microbial abundance
declined with depth more strongly in Hütelmoor than in Zarnekow (Fig. 7). There was a
pronounced decrease in microbial abundances at 20 cm depth in the Hütelmoor. For example, 16S
rRNA gene and *pmoA* gene copy numbers in deeper samples (below 20 cm depth) are one order
of magnitude lower than in upper samples on average, while the *mcrA* gene abundance are
approximately two orders of magnitude lower. Hütelmoor samples also exhibited larger
heterogeneity in terms of abundances than Zarnekow samples.

**4 Discussion**
**4.1 Fen geochemistry and relations to microbial community composition**
The rewetting of drained fens promotes elevated $CH_4$ production and emission, which can
potentially offset carbon sink benefits. Very few studies have attempted to link microbial
community dynamics and site geochemistry with observed patterns in $CH_4$ production and/or
emission in rewetted fens while such data are crucial for predicting long-term changes to $CH_4$
cycling (Galand et al. 2002, Yrjälä et al. 2011, Juottonen et al. 2012). In this study, we show that
$CH_4$-cycling microbial community composition is related to patterns in site geochemistry in two



rewetted fens with high $CH_4$ emissions, high methanogen abundances, and low methanotroph
abundances. Our results suggest that high methanogen abundances concurrent with low
methanotroph abundances contribute to increased $CH_4$ production and the resulting high emissions
in rewetted peatlands with readily available substrate. Thus, we present microbial evidence for
sustained elevated $CH_4$ emissions in mostly inundated rewetted temperate fens.
The environmental conditions and associated geochemistry of the two rewetted fens were largely
different. Depth profiles of porewater geochemical parameters show the fens differed in EC
throughout the entire peat profile, while pH and concentrations of alternative TEAs differed at
certain depths. In general, concentrations of TEAs oxygen, sulfate, nitrate, and iron were higher
in the Hütelmoor. In Zarnekow, geochemical conditions varied little across the fen and along the
peat depth profiles (Fig. 5). As expected, the geochemical heterogeneity was reflected in microbial
community structure in both sites, suggesting the importance of environmental characteristics and
associated geochemical conditions as drivers of microbial community composition (Figs. 2, 3, 4,
6). The NMDS ordinations (Fig. 6) show significant variation in archaeal and bacterial community
composition in the coastal brackish fen, and much less variation in the freshwater riparian fen.
Environmental fit vectors (Fig. 6) suggest that salinity (indicated by the EC vector), pH, oxygen
and alternative TEA availability are the most important measured factors influencing microbial
communities in the two fens. Patterns in microbial community composition have previously been
linked to salinity (e.g., Chambers et al. 2016), pH (e.g., Yrjälä et al. 2011), and TEA availability
in peatlands (e.g., He et al. 2015).
Comparing the geochemical depth profiles (Fig. 5) with the relative abundance of bacteria and
archaea (Figs. 3 and 4) provides a more complete picture of the relationships between microbial
communities and site geochemistry, particularly with respect to TEA utilization. While the
porewater depth profiles suggest there is little nitrate available for microbial use in HC 1, the
relative abundance plot for Archaea showed that this core was dominated by ANME-2D. ANME-
2D were recently discovered to be anaerobic methanotrophs that oxidize $CH_4$ performing reverse





methanogenesis using nitrate as an electron acceptor (Haroon et al. 2013). However, ANME-2D
has also been implicated in the iron-mediated anaerobic oxidation of methane (Ettwig et al. 2016),
and the HC 1 site showed slightly higher total iron concentrations. The relevance of ANME-2D as
$CH_4$ oxidizers in terrestrial habitats is still not clear. Rewetting converts the fens into widely
anaerobic conditions, thus providing conditions suitable for the establishment of anaerobic
oxidation of methane, but this has yet to be demonstrated in fens. The patchy occurrence and
locally high abundance of ANME-2D both in Hütelmoor and in Zarnekow suggests an ecological
relevance of this group. Shifts toward a less negative $\delta^{13}C$-$CH_4$ signature in the upper peat profile,
especially in HC 1 where ANME-2D was abundant, may indicate partial oxidation occurred, but
we could only speculate whether or not they are actively involved in $CH_4$ oxidation.
Although TEA input may be higher in the Hütelmoor, here, methanogenic conditions also
predominate. This finding contrasts the measured oxygen concentrations in the upper peat profile,
however seasonal analysis of oxygen concentrations in both sites suggests highly fluctuating
oxygen regimes both spatially and temporary (data not shown). Such non-uniform distribution of
redox processes has already been described elsewhere, in particular for methanogenesis (Hoehler
et al. 2001, Knorr et al. 2009). It is possible that oxygen levels in both fens are highly dynamic
allowing for both aerobic and anaerobic carbon turnover processes. Further, oxygen may not
necessarily be available within aggregates in which anaerobic pathways predominate. Anaerobic
conditions are also reflected by the extensive and stable occurrence of the strictly anaerobic
syntrophs (e.g., *Syntrophobacteraceae*, *Syntrophaceae*) in most samples, even in the top
centimeters. This suggests that syntrophic degradation of organic material is taking place in the
uppermost layer and the fermented substances are easily available for methanogens. Recent studies
from wetlands also show that methanogenesis can occur in aerobic layers, driven mainly by
Methanosaeta (Narrowe et al. 2017, Wagner 2017), which were detected in a high abundance in
this study (Fig. 4). As geochemistry and microbial community composition differ among the sites
in this study, it is thus notable that a similarly high abundance of methanogens, and low abundance





of methanotrophs was detected in both fens. The dominance of methanogens implies that readily
available substrates and favorable geochemical conditions promote high anaerobic carbon turnover
despite seasonally fluctuating oxygen concentrations in the upper peat layer.
**4.2 Microbial evidence for high CH$_4$ emissions**
Methanogens (mainly *Methanosaetaceae*) dominated nearly all of the various niches detected in
this study, while methanotrophs were highly under-represented in both sites (Figs. 3 and 4).
Functional and ribosomal gene copy numbers not only show a high ratio of methanogen to
methanotroph abundance (Fig. 7) irrespective of site and time of sampling, but also a small
contribution of methanotrophs to total bacterial population in both sites. Methanotrophs constitute
only ~0.06% of the total bacterial population in the Hütelmoor and ~0.05% at Zarnekow. It should
be noted that in this study we measured only gene abundances and not transcript abundances, so
that the pool both of active methanogens and methanotrophs was likely smaller than the numbers
presented here (Freitag and Prosser 2009, Freitag et al. 2010, Cheema et al. 2015, Franchini et al.
2015). Also, as we were unable to obtain microbial samples from before rewetting, a direct
comparison of microbial abundances was not possible. Compared to pristine fens, however, we
detected a relatively low abundance of methanotrophs. Liebner et al. (2015), for example, found
methanotrophs represented 0.5% of the total bacterial community in a pristine, subarctic
transitional bog/fen palsa, while *mcrA* and *pmoA* abundances were nearly identical. In a pristine
Swiss alpine fen, Liebner et al. (2012) found methanotrophs generally outnumbered methanogens
by an order of magnitude. Cheema et al. (2015) and Franchini et al. (2015) reported *mcrA*
abundances higher than *pmoA* abundances by only one order of magnitude in a separate Swiss
alpine fen. In the rewetted fens in our study, *mcrA* gene abundance was up to two orders of
magnitude higher than *pmoA* abundance (Fig. 7). As most methanotrophs live along the oxic-
anoxic boundary of the peat surface and plant roots therein (Le Mer and Roger 2001), the low
methanotroph abundances in both fens could be explained by disturbances to this boundary zone
and associated geochemical pathways following inundation. In rewetted fens, a massive plant



dieback has been observed along with strong changes in surface peat geochemistry (Hahn-Schöfl
et al. 2011, Hahn et al. 2015). The anoxic conditions at the peat surface caused by inundation may
have disturbed existing methanotrophic niches, and further, hindered the establishment of new
ones, as oxygen availability is the most important factor governing the activity of most
methanotrophs (Le Mer and Roger 2001, Hernandez et al. 2015).
Comparable studies have so far been conducted in nutrient-poor or mesotrophic fens where post-
rewetting $CH_4$ emissions, though higher than pre-rewetting, did not exceed those of similar pristine
sites (e.g., Yrjälä et al. 2011, Juottonen et al. 2005, Juottonen et al. 2012). Nevertheless, there is
mounting evidence linking $CH_4$-cycling microbe abundances to $CH_4$ dynamics in rewetted fens.
Juottonen et al. (2012), for example, compared *pmoA* gene abundances in three natural and three
rewetted fens and found them to be lower in rewetted sites. The same study also measured a lower
abundance of *mcrA* genes in rewetted sites, which was attributed to a lack of available labile carbon
compounds. In peatlands, and especially fens, litter and root exudates from vascular plants can
stimulate $CH_4$ emissions (Megonigal et al. 2005, Bridgham et al. 2013, Agethen and Knorr 2018),
and excess labile substrate has been proposed as one reason for dramatic increases in $CH_4$
emissions in rewetted fens (Hahn-Schöfl et al. 2011). Future studies should compare pre- and post-
rewetting microbial abundances along with changes in $CH_4$ emissions, plant communities, and
peat geochemistry to better assess the effect rewetting has on the $CH_4$-cycling microbial
community.

**5 Conclusion**
Despite a recent increase in the number of rewetting projects in Northern Europe, few studies have
characterized $CH_4$-cycling microbes in restored peatlands, especially fens. In this study, we show
that rewetted fens differing in geochemical conditions and microbial community composition have
a similarly low abundance of methanotrophs, a high abundance of methanogens, and an established
anaerobic carbon cycling microbial community. Comparing these data to pristine wetlands with

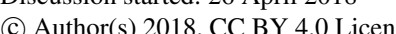



lower CH$_4$ emission rates, we found that pristine wetlands generally have a higher abundance of
methanotrophs than measured in the fens in this study, suggesting the inundation and associated
anoxia caused by flooding disturbs methanotrophic niches and may negatively affect the ability of
methanotrophic communities to establish. The abundances of methane producers and consumers
are thus suggested as important drivers for continued elevated CH$_4$ emissions following the
rewetting of drained fens. Our results suggest that in the context of CH$_4$ cycling, rewetting drained
peatlands by flooding may be problematic if post-rewetting conditions hinder methanotroph
establishment. Management decisions regarding rewetting processes should consider that
disturbances to methanotrophic niches is possible if rewetting leads to long-term inundation of the
peat surface.

**Competing interests**
The authors declare that they have no conflict of interest.

**6 Acknowledgements**
This study was conducted within the framework of the Research Training Group 'Baltic
TRANSCOAST' funded by the DFG (Deutsche Forschungsgemeinschaft) under grant number
GRK 2000. This is Baltic TRANSCOAST publication no. GRK2000/000X. The financial support
to Xi Wen (Grant No. 201408620031 to X.W.) provided by the China Scholarship Council (CSC)
is gratefully acknowledged. This study was supported by the Helmholtz Gemeinschaft (HGF) by
funding the Helmholtz Young Investigators Group of S.L. (VH-NG-919) and T.S. (Grant VH-NG-
821), a Helmholtz Postdoc Programme grant to F.K. (Grant PD-129), and further supported by the
Terrestrial Environmental Observatories (TERENO) Network. The Leibniz Institute for Baltic Sea
Research (IOW) is also acknowledged for funding the lab work in this study. The European Social
Fund (ESF) and the Ministry of Education, Science and Culture of Mecklenburg-Western
Pomerania funded this work within the scope of the project WETSCAPES (ESF/14-BM-A55-



0030/16). Dr. Matthias Gehre, head of the Laboratory of Stable Isotopes at the Helmholtz Centre
for Environmental Research, is acknowledged for measuring carbon isotopes of methane from
Hütelmoor samples. Anke Saborowski and Anne Köhler are also acknowledged for support in the
laboratory.















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





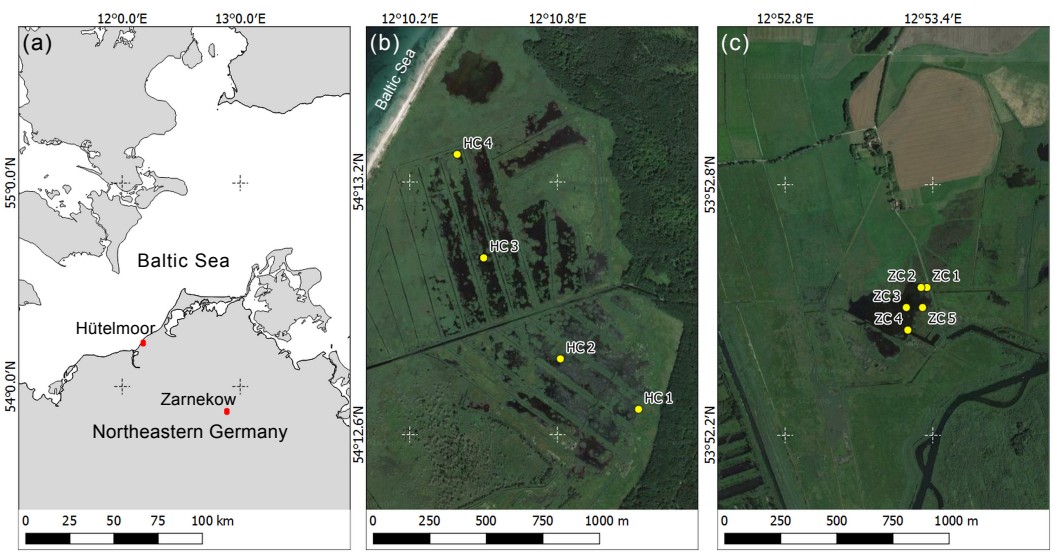


**Figure 1:** Location of study sites in northeastern Germany (a) and sampling locations within sites (b) Hütelmoor and (c) Zarnekow. Maps b) and c) are drawn to the same scale. Image source: (a) QGIS, (b) and (c) Google Earth via QGIS OpenLayer Plugin. Imagery date: August 9, 2015.






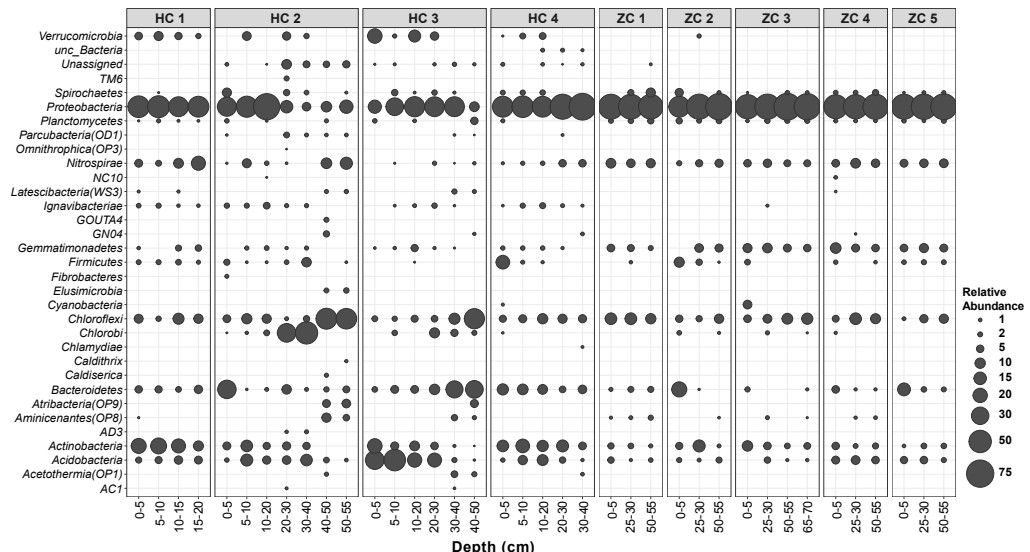

**Figure 2:** Relative abundances of different bacterial lineages in the study sites. Along the horizontal axis samples are arranged
according to site and depth. The rank order along the vertical axis is shown for the phylum level.



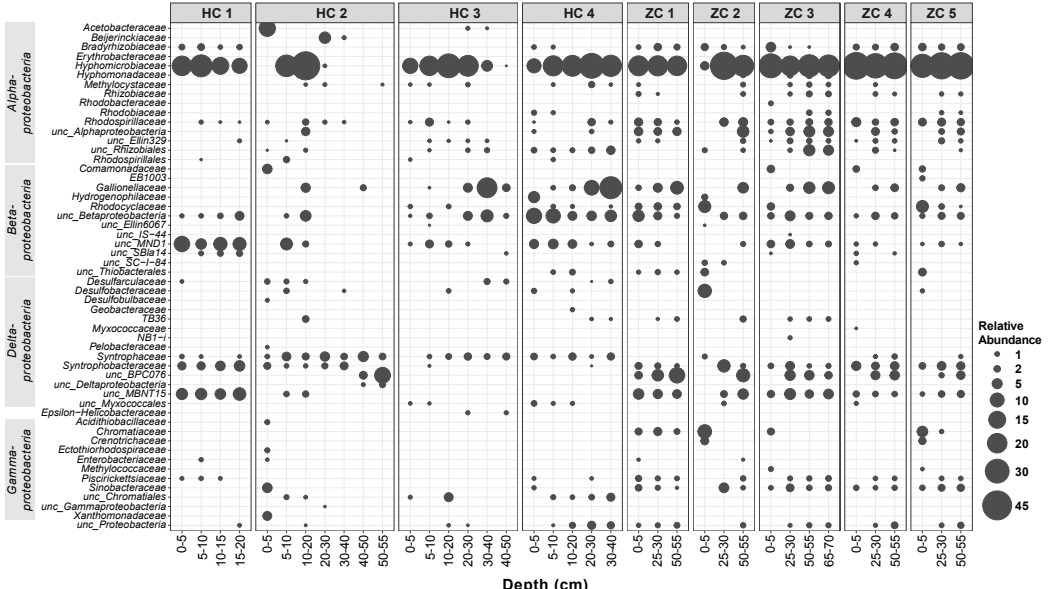

**Figure 3:** Relative abundances of Proteobacteria phyla in the study sites. Along the horizontal axis samples are arranged according to site and depth. The rank order along the vertical axis is shown for the family level. If an assignment to the family level was not possible the next higher assignable taxonomic level was used.




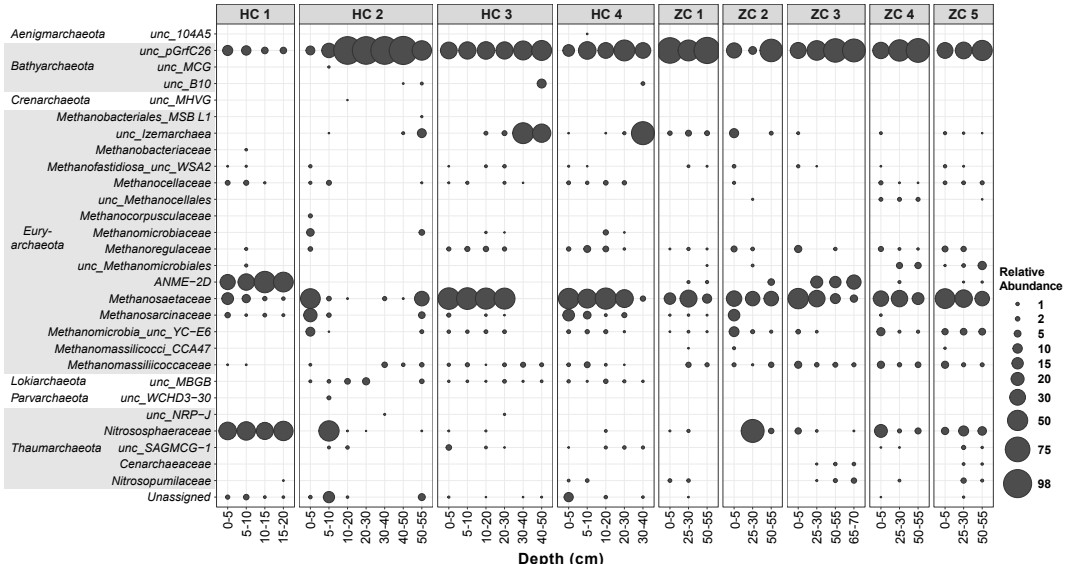

**Figure 4:** Relative abundances of different archaeal lineages in the study sites. Along the horizontal axis samples are arranged
according to site and depth. The rank order along the vertical axis is shown for the family level. If an assignment to the family level
was not possible, the next higher assignable taxonomic level was used.






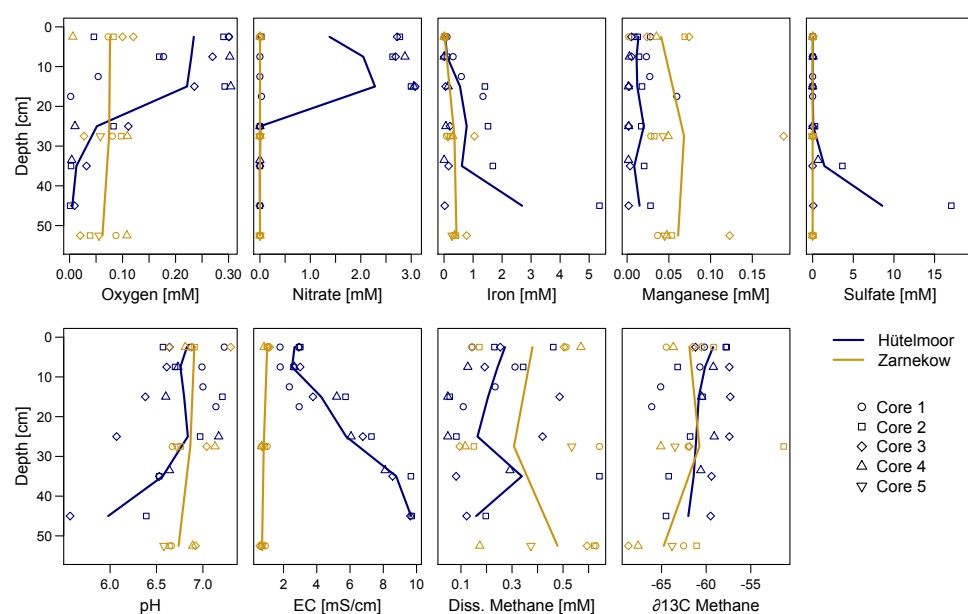

**Figure 5:** Depth profiles of porewater geochemistry (see x-axis labels for considered variables) in both study sites. Lines connect
the respective means.

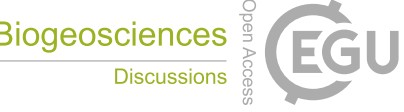




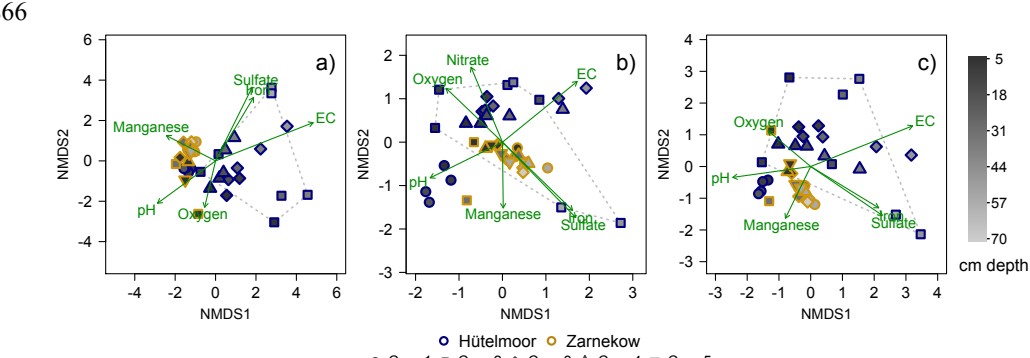

**Figure 6:** NMDS plots showing (a) bacterial, (b) archaeal, and (c) microbial (bacterial plus archaeal) community composition across the nine peat cores and their respective depth sections. The point positions represent distinct microbial communities, with the border colors of the symbols referring to the study sites and their shapes representing the core number. The shading indicates sample depth, with darker shades representing shallower depths, and lighter shades representing deeper depths. The dashed grey polygon highlights the large variation in microbial community composition in HC 2. Environmental fit vectors with a significance of $p < 0.05$ are shown in green.






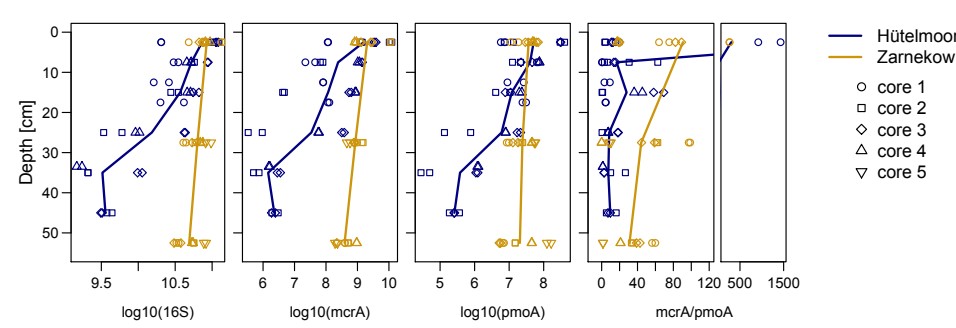

**Figure 7:** Depth distribution of qPCR abundances for total microbial (16S), methanogen (*mcrA*), methanotroph (*pmoA*), and ratio
of *mcrA* to *pmoA* gene copy numbers in both sites. Microbial abundances were designated as numbers of gene copies per gram of
dry peat soil and are shown against sampling depth using log-transformed values. Solid lines indicate mean abundances. Note that
the plot at the right was split into two plots to capture very high *mcrA/pmoA* ratios in the upper peat layer.





**Table 1:** Environmental conditions, geochemical conditions, and microbial abundances in peat cores from the Hütelmoor, a coastal minerotrophic fen
in northeastern Germany. Environmental conditions are described by pH and EC (electrical conductivity). Geochemical parameters shown are dissolved
methane ($CH_4$) concentrations, the isotopic signature of methane-bound carbon ($\partial^{13}C$–$CH_4$), and concentrations of terminal electron acceptors which
are denoted with their respective chemical abbreviations. Microbial abundances here represent the mean value of averaged subsamples for each depth
section (n=2). nd = not detected.

| Core, depth | pH | EC | $\partial^{13}C$–$CH_4$ | Dissolved $CH_4$ | $O_2$ | $NO_3^-$ | Fe | Mn | $SO_4^{2-}$ | 16S | *mcrA* | *pmoA* | *mcrA/pmoA* |
|---|---|---|---|---|---|---|---|---|---|---|---|---|---|
| cm | | mS cm$^{-1}$ | | | | mM | | | | gene copies g dry peat$^{-1}$ | | | |
| **HC 1**, 0–5 | 7.2 | 1.79 | -60.2 | 0.14 | 0.30 | nd | 0.10 | 0.03 | 0.03 | $2.04\times10^{10}$ | $1.15\times10^{08}$ | $6.60\times10^{06}$ | 17.7 |
| 5–10 | 7.0 | 1.80 | -60.7 | 0.31 | 0.18 | nd | 0.31 | 0.02 | 0.01 | $3.25\times10^{10}$ | $3.36\times10^{07}$ | $6.68\times10^{07}$ | 0.51 |
| 10–15 | 7.0 | 2.35 | -65.1 | 0.23 | 0.05 | nd | 0.60 | 0.03 | nd | $2.11\times10^{10}$ | $8.12\times10^{07}$ | $1.76\times10^{07}$ | 6.12 |
| 15–20 | 7.1 | 2.94 | -66.1 | 0.11 | nd | 0.03 | 1.34 | 0.06 | nd | $3.08\times10^{10}$ | $1.21\times10^{08}$ | $2.76\times10^{07}$ | 4.41 |
| **HC 2**, 0–5 | 6.9 | 3.01 | -57.8 | 0.46 | 0.05 | 0.03 | 0.03 | 0.01 | nd | $1.10\times10^{11}$ | $1.13\times10^{10}$ | $1.03\times10^{07}$ | 1,170 |
| 5–10 | 6.7 | 2.60 | -63.2 | 0.34 | 0.17 | 2.63 | 0.10 | 0.01 | 0.01 | $5.51\times10^{10}$ | $7.27\times10^{07}$ | $1.69\times10^{07}$ | 4.73 |
| 10–20 | 7.2 | 5.73 | -60.4 | 0.06 | 0.29 | 3.00 | 1.41 | 0.02 | nd | $3.13\times10^{10}$ | $4.47\times10^{06}$ | $7.32\times10^{06}$ | 0.74 |
| 20–30 | 7.0 | 7.29 | -61.8 | 0.08 | 0.08 | nd | 1.51 | 0.02 | 0.29 | $4.71\times10^{09}$ | $6.41\times10^{05}$ | $4.50\times10^{05}$ | 3.75 |
| 30–40 | 6.5 | 9.66 | -64.2 | 0.64 | nd | nd | 1.68 | 0.02 | 3.66 | $2.09\times10^{09}$ | $6.21\times10^{05}$ | $3.90\times10^{04}$ | 18.3 |
| 40–50 | 6.4 | 9.71 | -64.5 | 0.20 | nd | nd | 5.35 | 0.03 | 17.1 | $4.09\times10^{09}$ | $2.47\times10^{06}$ | $2.75\times10^{05}$ | 10.7 |
| **HC 3**, 0–5 | 6.6 | 2.93 | -57.7 | 0.23 | 0.29 | 2.77 | 0.11 | 0.01 | 0.04 | $1.10\times10^{11}$ | $1.34\times10^{09}$ | $3.51\times10^{08}$ | 3.86 |
| 5–10 | 6.6 | 3.00 | -57.4 | 0.19 | 0.27 | 2.69 | 0.01 | 0.01 | 0.03 | $8.72\times10^{10}$ | $1.40\times10^{09}$ | $3.42\times10^{07}$ | 46.6 |
| 10–20 | 6.4 | 3.77 | -57.3 | 0.49 | 0.24 | 3.08 | 0.05 | nd | nd | $6.08\times10^{10}$ | $5.86\times10^{08}$ | $9.35\times10^{06}$ | 63.6 |
| 20–30 | 6.1 | 6.77 | -57.4 | 0.42 | 0.11 | nd | 0.20 | nd | nd | $4.26\times10^{10}$ | $3.48\times10^{08}$ | $1.92\times10^{07}$ | 18.2 |
| 30–40 | 6.5 | 8.56 | -59.4 | 0.08 | 0.03 | nd | 0.16 | nd | nd | $1.05\times10^{10}$ | $3.20\times10^{06}$ | $1.17\times10^{06}$ | 2.74 |
| 40–50 | 5.6 | 9.36 | -59.5 | 0.12 | 0.01 | nd | 0.02 | nd | 0.08 | $3.18\times10^{09}$ | $2.16\times10^{06}$ | $2.58\times10^{05}$ | 8.39 |
| **HC 4**, 0–5 | 6.6 | 2.93 | -61.2 | 0.25 | 0.30 | 2.72 | 0.02 | 0.01 | 0.04 | $1.17\times10^{11}$ | $3.63\times10^{09}$ | $3.09\times10^{08}$ | 11.7 |
| 5–10 | 6.7 | 2.65 | -59.2 | 0.13 | 0.30 | 2.87 | 0.01 | nd | 0.05 | $4.87\times10^{10}$ | $1.09\times10^{09}$ | $7.51\times10^{07}$ | 14.5 |
| 10–20 | 6.6 | 5.20 | -60.5 | 0.05 | 0.30 | 3.05 | 0.14 | nd | nd | $4.85\times10^{10}$ | $8.71\times10^{08}$ | $2.15\times10^{07}$ | 40.8 |
| 20–30 | 7.2 | 6.06 | -59.1 | 0.05 | 0.01 | nd | 0.06 | nd | 0.02 | $9.78\times10^{09}$ | $5.82\times10^{07}$ | $7.91\times10^{06}$ | 7.36 |
| 30–40 | 6.6 | 8.11 | -60.6 | 0.29 | nd | nd | 0.09 | nd | 0.67 | $1.60\times10^{09}$ | $1.58\times10^{06}$ | $1.25\times10^{06}$ | 1.27 |



**Table 2:** Environmental conditions, geochemical conditions, and microbial abundances in peat cores from Zarnekow, a freshwater minerotrophic fen
in northeastern Germany. Environmental conditions are described by pH and EC (electrical conductivity). Geochemical parameters shown are dissolved
methane ($CH_4$) concentrations, the isotopic signature of methane-bound carbon ($\partial^{13}C$–$CH_4$), and concentrations of terminal electron acceptors which
are denoted with their respective chemical abbreviations. Microbial abundances here represent the mean value of averaged subsamples for each depth
section (n=2). nd = not detected.

| Core, depth | pH | EC | $\partial^{13}C$–$CH_4$ | Dissolved $CH_4$ | $O_2$ | $NO_3^-$ | Fe | Mn | $SO_4^{2-}$ | 16S | *mcrA* | *pmoA* | *mcrA/pmoA* |
|---|---|---|---|---|---|---|---|---|---|---|---|---|---|
| cm | | mS cm⁻¹ | | | | mM | | | | | | gene copies g dry peat⁻¹ | |
| **ZC 1**, 0–5 | 6.64 | 1.03 | -64.5 | 0.51 | 0.07 | 0.001 | 0.007 | 0.002 | 0.002 | $6.33 \times 10^{10}$ | $1.02 \times 10^{09}$ | $1.49 \times 10^{07}$ | 69.7 |
| 25–30 | 6.67 | 1.14 | -62.0 | 0.64 | 0.08 | 0.001 | 0.087 | 0.028 | 0.003 | $4.25 \times 10^{10}$ | $8.96 \times 10^{08}$ | $9.14 \times 10^{06}$ | 98.0 |
| 50–55 | 6.66 | 1.31 | -62.5 | 0.63 | 0.09 | 0.005 | 0.310 | 0.037 | 0.002 | $3.40 \times 10^{10}$ | $3.97 \times 10^{08}$ | $6.85 \times 10^{06}$ | 58.1 |
| **ZC 2**, 0–5 | 6.91 | 1.00 | -59.2 | 0.17 | 0.08 | 0.004 | 0.012 | 0.069 | 0.007 | $1.43 \times 10^{11}$ | $1.14 \times 10^{10}$ | $4.35 \times 10^{07}$ | 261 |
| 25–30 | 6.76 | 1.29 | -51.3 | 0.15 | 0.10 | 0.001 | 0.215 | 0.033 | 0.013 | $6.44 \times 10^{10}$ | $1.45 \times 10^{09}$ | $2.34 \times 10^{07}$ | 61.8 |
| 50–55 | 6.64 | 1.52 | -61.1 | 0.62 | 0.04 | nd | 0.410 | 0.054 | 0.003 | $5.64 \times 10^{10}$ | $5.10 \times 10^{08}$ | $1.50 \times 10^{07}$ | 34.0 |
| **ZC 3**, 0–5 | 6.88 | 1.17 | -60.5 | 0.50 | 0.10 | 0.001 | 0.073 | 0.074 | 0.032 | $7.86 \times 10^{10}$ | $2.78 \times 10^{09}$ | $3.26 \times 10^{07}$ | 85.7 |
| 25–30 | 7.04 | 3.39 | -61.9 | 0.10 | 0.03 | 0.002 | 1.046 | 0.188 | 0.003 | $5.79 \times 10^{10}$ | $7.81 \times 10^{08}$ | $1.55 \times 10^{07}$ | 51.8 |
| 50–55 | 6.92 | 3.82 | -68.7 | 0.59 | 0.02 | nd | 0.779 | 0.123 | 0.003 | $3.41 \times 10^{10}$ | $2.21 \times 10^{08}$ | $5.41 \times 10^{06}$ | 40.9 |
| **ZC 4**, 0–5 | 7.3 | 1.06 | -61.5 | 0.14 | 0.12 | 0.010 | 0.013 | 0.024 | 0.035 | $7.19 \times 10^{10}$ | $1.28 \times 10^{09}$ | $6.53 \times 10^{07}$ | 19.6 |
| 25–30 | 7.13 | 1.58 | -65.1 | 0.12 | 0.11 | 0.002 | 0.301 | 0.049 | 0.002 | $7.19 \times 10^{10}$ | nd | $4.60 \times 10^{07}$ | - |
| 50–55 | 6.89 | 1.51 | -67.6 | 0.17 | 0.11 | 0.002 | 0.366 | 0.048 | 0.002 | $5.42 \times 10^{10}$ | $9.47 \times 10^{08}$ | $4.50 \times 10^{07}$ | 21.0 |
| **ZC 5**, 0–5 | 6.81 | 0.83 | -63.7 | 0.57 | 0.01 | 0.002 | 0.005 | 0.035 | 0.005 | $8.73 \times 10^{10}$ | $8.73 \times 10^{08}$ | $4.97 \times 10^{07}$ | 17.6 |
| 25–30 | 6.72 | 0.86 | -63.5 | 0.53 | 0.06 | 0.002 | 0.139 | 0.043 | 0.001 | $8.94 \times 10^{10}$ | $5.21 \times 10^{08}$ | $5.57 \times 10^{07}$ | 93.4 |
| 50–55 | 6.58 | 1.00 | -63.8 | 0.37 | 0.06 | 0.002 | 0.275 | 0.045 | 0.002 | $8.00 \times 10^{10}$ | $2.14 \times 10^{08}$ | $1.44 \times 10^{08}$ | 14.9 |
