# Peer review of "Predominance of methanogens over methanotrophs in rewetted"

_Biogeosciences, 2018_

## Referee Comment (RC1) · Anonymous Referee #1 · 14 May 2018

Review of the Biogeosciences Discuss. Paper "Predominance of methanogens over methanotrophs contributes to high methane emissions in rewetted fens by Wen et al. The authors present high throughput sequencing and qPCR data of microbial communities of two rewetted fens in northern Germany. Next to the microbial analyses the pore water chemistry, dissolved methane and the isotopic signal of the methane C was analyzed. The paper is well written but hampers in the experimental design and some missing analyses. First of all there are no datasets or samples available which connect the rewetting treatment to a control or a pre-disturbance measurement. With pre-disturbance we can argue the existence of the drained fen performance or even the performance of the fen before drainage. So to what can the results be compared?

[Figure]

I thought that the lateral scanning of the fens by different sampling points could explain this but actually the data is not discussed in this sense. At least at Huettelmoor the gradient goes away from the dam. The data is also not discussed to the methane fluxes of the different sampling points. For Huettelmoor they exist because there have been chamber measurements which should match quite close to the H1-4 cores if they are not exactly at the same spot. I also wonder why no potential activity measurement was performed to assess the activity of methane production and oxidation. Can this still be performed because it would give much information which is not told by the community analyses of gene copy numbers. Next I miss in the qPCR approach the measurement for Archaea. Why has this not be measured. In the MM section the authors should tell which depths have been sampled at each site. I can see the depths in the Figs BUT they need to be told in the MM. You have also to discuss in the Ms why the depth sampling was so different between fens and within the Huettelmoor fen. Probably the fens were never mentioned to be published together otherwise the sampling would be convergent. In the MM section I miss the sample n AND I want to point out that you have not replicated your study design. In my opinion this is a harsh critique. Taking two within replicates for DNA extraction is not the same you should have two to three adjacent lines. In the Intro and Discussion it is stressed that elevated methane emissions after rewetting is dangerous. I doubt that. First the dried peatland lost a lot of CO2 due to peat degradation and the onset of methane emission after restoration is a hint that peat formation starts to accelerate again and this process fixes more C than it loses. There is scientific literature around this and you may bring this into your discussion. In the MM I do not see if the rewetted Huettelmoor water table is 0.6 m above or below peat surface (line 126). In the Results of the MM statistical chapter I miss information of have many sequences were retrieved. How many OTUs were obtained and the bubble data is generated on and how many observations. In lines 201-202 is something I do not understand. Three PCR products of the same sample were combined. OK but why. But the next sentence says PCR products of different samples were pooled. . .???On lines 273-276 give the percentage of Methanotrophs

out of the total. You tell them in the discussion. This so, because you present for Methanogens this data on line 280. Looking at the Figs you have no real depth separation in your measured variables at Zarnekow. WHY? For the end; line 78-80 states wrong: there are more publications to the theme Reumer et al. 2018. Impact of peat mining, and restoration on methane turnover potentials and methane-cycling microorganisms in a northern bog. Applied and Environmental Microbiology 84, 3 e02218-17. https://doi.org/10.1128/AEM.02218-17. Putkinen et al. 2018. Recovery of methane turnover and the associated microbial communities in restored cut-away peatlands is strongly linked with increasing Sphagnum abundance. Soil Biology & Biochemistry 116: 110-119.

---

## Referee Comment (RC2) · Anonymous Referee #2 · 15 May 2018

[revised manuscript text omitted]

---

## Referee Comment (RC3) · Anonymous Referee #3 · 15 May 2018

Wen et al. address microbial controls of high methane emission after re-wetting in two temperate peatlands with contrasting geochemistry. There is very little information available on microbiology of re-wetted peatlands, so as the first study of re-wetted non-acidic fens, this study is very welcome. The manuscript is clearly written and easy to follow. The molecular analyses for microbes have been carried out with care (testing for sample inhibition in qPCR, pooling three different PCR products to reduce amplification bias, checking the taxonomic affiliations of OTUs in ARB). This is not a study of rewetting effects, because no samples from before re-wetting or from a non-rewetted control site are available. However, in addition to providing much needed information on re-wetted peatlands, the results contain some interesting details such

as the strikingly patchy distribution of ANME-2d.

My biggest concern is that the main result is based on comparison of two different qPCR assays (mcrA vs. pmoA). Such a direct comparison of values assumes nearly absolute quantification, which is not realistic for environmental samples (different limitations in coverage for each primer pair etc). Comparisons of values of one assay between samples, on the other hand, do not rely on this assumption in the same way. The previous examples of pristine wetlands used as support (l. 413-421, 450-452) similarly rely on comparisons of two different qPCR assays. If/when these studies have used different methods and primers as this study, the comparisons become even more problematic, even when made at the broad level of orders of magnitude. I do not disagree with the overall conclusion that high numbers of methanogens the most likely reason for the high methane fluxes, but I would strongly recommend addressing this limitation in the discussion and modifying the text on l. 404-421 and elsewhere, including the title of the manuscript. Maybe strengthening the interpretation of microbial community results in relation to geochemistry could provide an alternative main message.

In addition, I am wondering about the role of methanotrophs in completely inundated peat and in the water layer. It is very much expected that methanotrophic activity would be low considering that in both sites the sampled peat was inundated. The optimal peat layer for methanotrophs where both methane and oxygen are readily available is largely missing (which the authors do address in the end of the manuscript). However, such conditions could be present in the water layer. I realise the water layer is out of the scope of this study, but are there reasons to exclude it from discussion or assume it plays no role in methane oxidation?

Minor comments:

1. l. 190-193 Did the primers contain sequencing adapters and barcodes or were they added later?

2. l. 234-235 Please remove the word 'all' from 'suitable for detecting all aerobic

methanotrophic Proteobacteria', or change to 'all known' or similar (we cannot assume to be able to detect the full diversity).

3. l. 318-319, l. 360-361 The Hütelmoor samples show higher within-site variation, but the samples were also taken much further apart from each other. Could this not explain the larger variation? On l. 360-361, the sentence could be understood to suggest the difference is due to brackish vs. freshwater.

4. l. 360 Please change 'significant' to another word because no statistical testing was carried out for differences of community composition.

5. l. 415 I do not think it is possible to compare PCR-based relative abundances between different studies, unless the studies used completely identical methods and equipment. Was this the case with Liebner et al. 2015?

―――――――――――――――――

---

## Author Comment (AC1) · 23 Jun 2018

We thank the reviewer for their helpful comments and constructive suggestions which have resulted in a much-improved manuscript overall. Below you can find our full response to each point raised.

Response to Anonymous Referee #1
Review of the Biogeosciences Discuss. Paper "Predominance of methanogens over methanotrophs contributes to high methane emissions in rewetted fens by Wen et al. The authors present high throughput sequencing and qPCR data of microbial communities of two rewetted fens in northern Germany. Next to the microbial analyses the pore water chemistry, dissolved methane and the isotopic signal of the methane C was analyzed. The paper is well written but hampers in the experimental design and some missing analyses. First of all there are no datasets or samples available which connect the rewetting treatment to a control or a pre-disturbance measurement. With pre-disturbance we can argue the existence of the drained fen performance or even the performance of the fen before drainage. So to what can the results be compared? I thought that the lateral scanning of the fens by different sampling points could explain this but actually the data is not discussed in this sense. At least at Huettelmoor the gradient goes away from the dam.

**We agree, it would indeed be nice, to have pre-rewetting data. However, there were no pre-rewetting peat samples available to compare our microbial data to. For this reason, we performed an extensive literature search comparing the published geochemical and microbial characteristics of drained versus rewetted fens to the fens in this study, and we are confident that is a valid approach for discussing post-rewetting conditions. Having pre-rewetting data to compare to is, unfortunately, very rare for temperate, restored fens; thus, we rather discuss the post-rewetting conditions of the two fens and highlight differences among drained versus rewetted fens using information that is published and available.**

The data is also not discussed to the methane fluxes of the different sampling points. For Huettelmoor they exist because there have been chamber measurements which should match quite close to the H1-4 cores if they are not exactly at the same spot. I also wonder why no potential activity measurement was performed to assess the activity of methane production and oxidation. Can this still be performed because it would give much information which is not told by the community analyses of gene copy numbers.

**You are correct our approach does not allow a detailed consideration of different sampling plots. However, our aim was not to present plot-scale interpretations but rather discuss the ongoing high methane fluxes on a larger scale. The location of available chamber measurements do not match the locations for the analysis presented here, thus we consider emissions on an ecosystem- rather than plot-level scale.**
**For this study we did not perform incubations to determine rates of production or oxidation. Methane production is indeed high for both fens and this can also be inferred from the persistently high dissolved methane concentrations we present in this study and from unpublished earlier anaerobic incubations. In more detail, incubations were performed with Zarnekow peat and these data have been added to the manuscript as supplemental information. We have added an additional author (Paul Bodelier) as he has provided us with the incubation data. The data show that besides methane production the**

**potential for efficient methane oxidation also exists. Incubations provide ideal conditions for the organisms, and thus overestimate actual *in situ* methane oxidation. Specifically, in methane oxidation incubations excess of oxygen is available for methanotrophs which opposes *in situ* conditions in both fens where methane oxidation is overprinted by other processes. Unfortunately, no methane oxidation incubation data are available for the Hütelmoor because earlier attempts to measure methane oxidation in this site have failed.**

Next I miss in the qPCR approach the measurement for Archaea. Why has this not been measured.

**In our study we were seeking for microbial controls for ongoing high emissions of methane in the two studied fens and consequently sought the ratio between methanotrophs and methanogens using qPCR. We further wanted to assess the relative contribution of both groups with regard to total bacteria and archaea and therefore performed deep sequencing using the Illumina platform. With this we could already answer our initial question. Seeking a final proof for our qPCR analysis we also quantified total bacteria with qPCR. The ratio of methanotrophs to total bacteria based on qPCR is very much in line with the sequencing results supporting the robustness of our qPCR assays. The quantification of total archaea using qPCR was thus not necessary for answering our initial questions. Finally, as the reviewer may be aware of, primer- or probe-based quantifications of total archaea targeting their 16S rRNA gene is often hampered through co-amplification of bacteria given the large sequence similarities. In summary, we refrained from qPCR data for the archaeal community since it does not add to the presentation of our major finding(s).**

In the MM section the authors should tell which depths have been sampled at each site. I can see the depths in the Figs BUT they need to be told in the MM.

**As suggested the sampling depths were added to the text in the materials and methods section in lines 161-165.**

You have also to discuss in the Ms why the depth sampling was so different between fens and within the Huettelmoor fen. Probably the fens were never mentioned to be published together otherwise the sampling would be convergent.

**As suggested, we have altered the text in lines 165-167 and 177-179 to explain why the sampling and depth resolution was different between the fens. We'd like to emphasize that the data were indeed collected with comparison of the two fens in mind. The reason for the difference in sampling depth is that previous studies from Zarnekow show that the peat stratigraphy is much less variable than the stratigraphy at the Hütelmoor. Difference in porewater sampling methods was due to accessibility and sampling difficulty: the permanent porewater dialysis samplers could not be installed at the sampling locations in the Hütelmoor.**

In the MM section I miss the sample n AND I want to point out that you have not replicated your study design. In my opinion this is a harsh critique. Taking two within replicates for DNA extraction is not the same you should have two to three adjacent lines.

**You are correct, sample n should be given. We have added the sample n to our revised manuscript in the materials and methods section in lines 159-160. With this study, our aim was not to argue for differences between the sampling points within the fens, but to seek differences and similarities among both fens. In this regard, we have four (n=4) and five replicates (n=5), respectively.**

In the Intro and Discussion it is stressed that elevated methane emissions after rewetting is dangerous. I doubt that. First the dried peatland lost a lot of CO2 due to peat degradation and the onset of methane emission after restoration is a hint that peat formation starts to accelerate again and this process fixes more C than it loses. There is scientific literature around this and you may bring this into your discussion.

**We did not intend to state that methane emissions after rewetting are "dangerous" but in the flooded peatlands we know they are elevated and also for a quite substantial duration. It was generally assumed in rewetting projects that peat methane production returns to near neutral levels within several years but flooded hypertrophic fens might behave differently. This question puzzled us for quite some time, now but the data we present here may solve a part of that puzzle in that one reason could be the disproportionately low abundance of methanotrophs compared to other microbes. We have adjusted the text in the introduction so it does not imply that methane emissions are solely a negative phenomenon but may only be temporary.**

In the MM I do not see if the rewetted Huettelmoor water table is 0.6 m above or below peat surface (line 126).

**As suggested, the text was adjusted in line 136 to indicate that water level was 0.6 m above the peat surface.**

In the Results of the MM statistical chapter I miss information of have many sequences were retrieved. How many OTUs were obtained and the bubble data is generated on and how many observations.

**For archaea, a total of 6844177 valid sequences were obtained, ranging from 60496 to 398660 in individual samples. These sequences were classified into 402 OTUs. Then the OTU table was collapsed at higher taxonomic level to generate the bubble plot. For bacteria, a total of 2586148 valid sequences were obtained, ranging from 22826 to 164916 in individual samples. These sequences were classified into 843 OTUs. Then the OTU table was collapsed at higher taxonomic level to generate the bubble plot. This information was added to the materials and methods section in lines 243-248.**

In lines 201-202 is something I do not understand. Three PCR products of the same sample were combined. OK but why. But the next sentence says PCR products of different samples were pooled...???

**The samples were pooled to reduce amplification bias. We adjusted the text in lines 220-221 so that it is clear why the samples were pooled.**

On lines 273-276 give the percentage of Methanotrophs out of the total. You tell them in the discussion. This so, because you present for Methanogens this data on line 280.

**As suggested, in our revised manuscript we added the specific methanotroph abundances to the results section in lines 296-298.**

Looking at the Figs you have no real depth separation in your measured variables at Zarnekow. WHY?

**As detailed above, we have provided an explanation for the different sampling depths and lower depth resolution in Zarnekow in the methods section. (lines 165-167 and 177-179)**

For the end; line 78-80 states wrong: there are more publications to the theme Reumer et al. 2018. Impact of peat mining, and restoration on methane turnover potentials and methane-cycling microorganisms in a northern bog. Applied and Environmental Microbiology 84, 3 e02218-17. https://doi.org/10.1128/AEM.02218-17. Putkinen et al. 2018. Recovery of methane turnover and the associated microbial communities in restored cut-away peatlands is strongly linked with increasing Sphagnum abundance. Soil Biology & Biochemistry 116: 110-119.

**Thank you for suggested citations. We have adjusted the text accordingly and added the reference to line 84.**

---

## Author Comment (AC2) · 23 Jun 2018

We thank the reviewer for their helpful comments and constructive suggestions which have resulted in a much-improved manuscript overall. Below you can find our full response to each point raised.

Response to Anonymous Referee #2

The study of Wen et al on the "Predominance of methanogens .... in rewetted fans" is very well written and presented in a clear way. In this study the abundance and community structure of methanogenic and methanotrophic microorganisms in two rewetted fans is related to geochemical parameters. However, the study has in my opinion to major drawbacks: In the title and within the text the authors refer to methane emissions of the two fens, however no data on methane emission are presented. With so many authors involved there certainly should be data on this important factor?? The relation / explanation how the environmental parameters influence the abundance or community structure of the methane related organisms is not convincing; it seems to be rather biased. I know it is not easy to explain microbial patterns with geochemical ones, but I would suggest a more serious statistic here.

**With regards to the first drawback mentioned by the reviewer, we agree that the paper would profit from including actual methane emission data. To this end, we have added the most recent published values for average methane flux rates for both fens to the revised manuscript to lines 104-108.**

**With regards to the second major drawback mentioned by the reviewer, we believe 2-dimensional non-metric multidimensional scaling (NMDS) is a robust statistical method. The environmental fit to the NMDS is a statistical approach based on a Monte Carlo permutation that shows which variables are significantly related to the community structure of the microorganisms. For this reason, we feel that additional statistics are not necessary to support our overall conclusions based on the NMDS. We performed correlation analyses on methanotroph abundance versus oxygen and dissolved methane concentrations, however the relationships were not significant. Nevertheless, we failed to mention this in the original manuscript and have adjusted the text in lines 362-363.**

Line 201 "PCR products of three individual runs per sample were combined." – why this?

**The PCR products were combined to reduce amplification bias. A short phrase was added to the manuscript to make this clear to the reader in lines 220-221.**

Line 292 "I suggest to start the results section with the geochemical description of the study site"

**Thank you for the suggestion. In an earlier version of the manuscript, we had the geochemical section first but decided later that the main message of the paper is supported better when the microbial data were presented first. We tried both approaches, and still came to this conclusion. So, we hope it is acceptable that we keep the order of the presentation of our results.**

Line 295 "as you refer later in the discussion to salinity, it would be nice to have these values converted to PSU, for comparison with other studies"

We agree that it would be better to have salinity values for comparability. However, for brackish waters the calculation is unreliable as salinity in low-salt waters is not well-defined. This is an issue that is unresolved among hydrogeologists and chemical oceanographers alike, thus conversions from EC to psu are generally not performed for brackish systems. We would therefore suggest that we present our original EC data, which is more scientifically sound, but provide the information needed for conversion from EC to psu for the reader (in the methods section).

Line 322-323 "I do not think that "depth" is a suitable parameter here. It should be seen as envelopping parameter which is characterized in itself by NO3, SO4, O2 ...Also it makes figure 6 rather confusing. Maybe you could try to do the analysis without "depth", by pooling all the data? also, the parameter "site" could be omitted....

We did not want to include depth as a parameter but rather as a proxy for other parameters. We think including depth is important as it may stand for a proxy for other parameters which were not measured in this study.  We further believe that site is an important parameter here as comparison of the two fens is a main point of the paper. The inclusion of site in the NMDS reinforces our findings that both geochemistry and microbial community composition were much more variable in the Hütelmoor than in Zarnekow.

Line 347 "where does the emission data come from? Are there any data available??"

Data on methane exchange was recorded in both fens by us and other colleagues with chambers and eddy covariance in the past and still today. Since we have no measurements that are directly associated with the core samplings and the porewater sampling used here, we first decided to go without $CH_4$ exchange data. As two reviewers have criticized this point, we have updated the manuscript to include the most recent publish values for methane fluxes from the two fens in lines 104-108.

Line 362 "thus CH4 concentration had no influence?? At least for the methanotrophs this should be an important factor. If not, this should at least be stated so"

You are right, a correlation between methanotrophs and CH4 concentrations is an important factor. According to our analyses CH4 concentrations did not correlate with methanotroph abundance nor with the abundance of other microbes. Because many studies have found it to be an important factor influencing methanotroph populations, we should have nevertheless mentioned that we found no correlation in our study. Therefore, we now describe this in lines 362-363.

Line 365 "for comparison it would be nice to have EC converted to salinity"

We agree that it would be better to have salinity values for comparability. However, for brackish waters the calculation is unreliable as salinity in low-salt waters is not well-defined. This is an issue that is unresolved among hydrogeologists and chemical oceanographers alike, thus conversions from EC to psu are generally not performed for brackish systems. We would therefore suggest that we present our original EC data, which

is more scientifically sound, but provide the information needed for conversion from EC to psu for the reader (in the methods section).

Line 380 "I do not see any significant shifts in the figure, but only a scatter of data ...."

**The sentence was adjusted in lines 406-409 to help the reader follow the specifically mentioned shift in the isotopic data of HC 1. The delta signature of HC 1 (open circles) shifts from ~-65 to ~60 which could be the result of oxidation processes.**

Line 401 "however, it is not clear to me, why the abundance of methanotrophs is so low. Shifting O2 regimes should be no problem, as this is often the case in other environments, tidal sediments..."

**The argument for the low abundance of methanotrophs was strengthened in the revised manuscript. Specifically, in lines 461-467 we suggest that competition for oxygen with heterotrophic organisms rather than fluctuations in oxygen are likely a reason for the low abundance of methanotrophs. In fact, our data support this as our bubble plot for bacteria shows hyphomicrobiaceae dominated the bacterial community, a family of which the large majority are aerobic heterotrophs.**

Line 402 "The heading is not suitable here, as you only discuss the low abundance of methantrophs here. Data on methane emissions would be helpful here...."

**The authors agree that the headline was not suitable. We thus changed the headline in line 429 to better represent the section – "Low methanotroph abundances in rewetted fens"**

Line 423 "but these disturbances in O2 regime would be also inhibitory for the methanogens on the other side...."

**Though recent studies show that methanosaeta, which was the most abundant methanogen in this study, thrive even in oxic layers, it is also likely additional factors are affecting the methanotroph populations that were not thoroughly discussed. In our revised manuscript we expand on the discussion regarding the absence of methanotrophs. Specifically, we suggest in lines 461-467 that competition by heterotrophs which also utilize oxygen may ultimately be preventing methanotroph establishment.**

Line 428 "what about methane availability??"

**Substrate (i.e. methane) availability has indeed been shown to correlate with methanotroph populations. We have added this to our revised manuscript in lines 362-363 as previously mentioned. As mentioned above, in our study methanotroph abundance and methane concentrations did not correlate, though. Also, methane concentrations in the pore water were high throughout all sites so the availability of methane is unlikely to constrain methanotroph abundance in the two rewetted fens of our study.**

---

## Author Comment (AC3) · 23 Jun 2018

We thank the reviewer for their helpful comments and constructive suggestions which have resulted in a much-improved manuscript overall. Below you can find our full response to each point raised.

Response to Anonymous Referee #3
Wen et al. address microbial controls of high methane emission after re-wetting in two temperate peatlands with contrasting geochemistry. There is very little information available on microbiology of re-wetted peatlands, so as the first study of re-wetted non-acidic fens, this study is very welcome. The manuscript is clearly written and easy to follow. The molecular analyses for microbes have been carried out with care (testing for sample inhibition in qPCR, pooling three different PCR products to reduce amplification bias, checking the taxonomic affiliations of OTUs in ARB). This is not a study of rewetting effects, because no samples from before re-wetting or from a non- rewetted control site are available. However, in addition to providing much needed information on re-wetted peatlands, the results contain some interesting details such as the strikingly patchy distribution of ANME-2d.
My biggest concern is that the main result is based on comparison of two different qPCR assays (mcrA vs. pmoA). Such a direct comparison of values assumes nearly absolute quantification, which is not realistic for environmental samples (different limitations in coverage for each primer pair etc). Comparisons of values of one assay between samples, on the other hand, do not rely on this assumption in the same way. The previous examples of pristine wetlands used as support (l. 413-421, 450-452) similarly rely on comparisons of two different qPCR assays. If/when these studies have used different methods and primers as this study, the comparisons become even more problematic, even when made at the broad level of orders of magnitude. I do not disagree with the overall conclusion that high numbers of methanogens the most likely reason for the high methane fluxes, **b**ut I would strongly recommend addressing this limitation in the discussion and modifying the text on l. 404-421 and elsewhere, including the title of the manuscript. Maybe strengthening the interpretation of microbial community results in relation to geochemistry could provide an alternative main message.

**Please find our reply to these concerns below.**

In addition, I am wondering about the role of methanotrophs in completely inundated peat and in the water layer. It is very much expected that methanotrophic activity would be low considering that in both sites the sampled peat was inundated. The optimal peat layer for methanotrophs where both methane and oxygen are readily available is largely missing (which the authors do address in the end of the manuscript). However, such conditions could be present in the water layer. I realise the water layer is out of the scope of this study, but are there reasons to exclude it from discussion or assume it plays no role in methane oxidation?

**It is indeed possible that oxidation may be occurring in the water column. It is true, however, that the water column was beyond the scope of this manuscript. Recent, preliminary data for Zarnekow show methanotrophs in high abundance associated with ceratophyllum in the water column (unpublished data, still in progress). Nevertheless, even if oxidation is occurring in the water column in these two sites it is clearly not significant enough to keep methane concentrations and emissions low as demonstrated by**

the flux data (added) in the revised manuscript. This is now mentioned in the discussion in lines 464-466.

Minor comments:
1. l. 190-193 Did the primers contain sequencing adapters and barcodes or were they added later?

**Yes, the primers contained barcodes. A phrase was added to the manuscript in line 212 to denote that the primers contained barcodes in the 5'-end.**

2. l. 234-235 Please remove the word 'all' from 'suitable for detecting all aerobic methanotrophic Proteobacteria' or change to 'all known' or similar (we cannot assume to be able to detect the full diversity).

**As suggested we have changed the text in line 259 to instead say "all *known* aerobic methanotrophic Proteobacteria".**

3. l. 318-319, l. 360-361 The Hütelmoor samples show higher within-site variation, but the samples were also taken much further apart from each other. Could this not explain the larger variation? On l. 360-361, the sentence could be understood to suggest the difference is due to brackish vs. freshwater.

**Though in the study the Zarnekow samples were taken closer together, we know from previous work at the site that there is indeed less variation across the Zarnekow peatland (e.g. Zak and Gelbrecht 2007). Thus, taking the cores further apart in Zarnekow in this study would not have resulted in greater variation in our measured variable.**

4. l. 360 Please change 'significant' to another word because no statistical testing was carried out for differences of community composition.

**The phrase 'significant variation' was changed to 'large variation' in line 386 in the revised version of the manuscript.**

5. l. 415 I do not think it is possible to compare PCR-based relative abundances between different studies, unless the studies used completely identical methods and equipment. Was this the case with Liebner et al. 2015?

**It is an inevitable limitation that the methods and equipment of different studies are not completely identical. This is not only a limitation of our work but a general issue for meta-studies. All comparisons regarding bacterial, methanotrophic and methanogenic abundance are based on universal primer combinations of the respective groups. The primers we used for the bacterial 16S rRNA and mcrA genes of this study are identical with the primers used in Liebner et al. 2015. With regards to pmoA, both studies used universal primer combinations including identical forward primers, but as a result of initial testing different reverse primers. Further, the same qPCR technology was used. In addition, we compared the ratio of methanogenic to methanotrophic abundances and the fraction of methanotrophs in relation to the total bacterial community based on two independent methods, namely qPCR and sequencing, instead of the direct methanogenic or**

methanotrophic abundances. This kind of 'normalization' mitigates the bias of different experiments and makes the results more reasonable and reliable. As suggested, we now discuss this potential limitation in our revised manuscript in lines 448-453.

Further, we have revised the title of our manuscript. Our revised manuscript title is as follows, "Predominance of methanogens over methanotrophs in rewetted fens - a possible explanation for the observed high methane emissions?".

---

## Author Response (AR2)

[revised manuscript text omitted]

**List of relevant changes made to the manuscript:**
-the term 16s rDNA was changed to 16s rRNA in lines 38-39
-the incubation methods were added to the methods section
-the sample n for the incubations was added to the methods section and the description of figure S1
-depth shading was removed from figure 6
-Hütelmoor core 2 was highlighted in red in figure 6
**Authors' responses to referee reports:**
Dear Editor, Dear Referees,
We once again thank you for the constructive feedback on the manuscript. Please find our responses to the
individual suggestions below in bold text.
Anonymous Referee #2
"The revision of the Ms "Predominance of methanogens over methanotrophs in rewetted fens characterized by
high methane emissions" has significantly improved the Ms. All of my suggestions have been incorporated. Only
with figure 6, the NMDS plot, I still have some comments:
NMDS may be a standard statistical method, however I still think that figure 6 is rather confusing.
It is evident that the samples from Zarnekow are different from Hütelmoor, and with a lower variability. However,
the shading of the different depths is not discernible in the plots and within the figure I cannot detect the different
depths.
If HC2 is so much different than the other samples I suggest to choose another color /symbol for it to make this
better visible."
**We understand that the depth shading in figure 6 is confusing and we have therefore removed depth**
**shading from the figure. We have further highlighted Hütelmoor core 2 (red color inside symbol borders),**
**as suggested, to emphasize its difference from all other cores.**
Anonymous Referee #3
"I think the changes by the authors have improved the manuscript and I have only two further minor comments:

1. Please describe in the methods how the incubation data in Fig. S1 was obtained (or add a reference to the
method) and mention somewhere what the n is in Fig. S1.
**Thank you for pointing this out. We have added the incubation methods to the methods section, as well as**
**described the sample n in both the methods and in the chart description.**
2. On line 39, 16S rDNA -> 16S rRNA"
**The term "16S rDNA" has been changed to "16S rRNA".**